# Learning to Generate Wasserstein Barycenters

## Abstract

Optimal transport is a notoriously difficult problem to solve numerically, with current approaches often remaining intractable for very large scale applications such as those encountered in machine learning. Wasserstein barycenters – the problem of finding measures in-between given input measures in the optimal transport sense – is even more computationally demanding. By training a deep convolutional neural network, we improve by a factor of 60 the computational speed of Wasserstein barycenters over the fastest state-of-the-art approach on the GPU, resulting in milliseconds computational times on $512 \times 512$ regular grids. We show that our network, trained on Wasserstein barycenters of pairs of measures, generalizes well to the problem of finding Wasserstein barycenters of more than two measures. We validate our approach on synthetic shapes generated via Constructive Solid Geometry as well as on the "Quick, Draw" sketches dataset.

## 1 Introduction

Optimal transport is becoming widespread in machine learning, but also in computer graphics, vision and many other disciplines. Its framework allows for comparing probability distributions, shapes or images, as well as producing interpolations of these data. As a result, it has been used in the context of machine learning as a loss for training neural networks (Arjovsky et al., 2017), as a manifold for dictionary learning (Schmitz et al., 2018), clustering (Mi et al., 2018) and metric learning applications (Heitz et al., 2019), as a way to sample an embedding (Liutkus et al., 2019) and transfer learning (Courty et al., 2014), and many other applications (see Sec. 2.3). However, despite recent progress in computational optimal transport, in many cases these applications have remained limited to small datasets due to the substantial computational cost of optimal transport, in terms of speed, but also memory.

We tackle the problem of efficiently computing Wasserstein barycenters of measures discretized on regular grids, a setting common to several of these machine learning applications. Wasserstein barycenters are interpolations of two or more probability distributions under optimal transport distances. As such, a common way to obtain them is to perform a minimization of a functional involving optimal transport distances or transport plans, which is thus a very costly process. Instead, we directly predict Wasserstein barycenters by training a Deep Convolutional Neural Network (DCNN) specific to this task.

An important challenge behind our work is to build an architecture that can handle a variable number of input measures with associated weights without needing to retrain a specific network. To achieve that, we specify and adapt an architecture designed for and trained with two input measures, and show that we can use this modified network with no retraining to compute barycenters of more than two measures. Directly predicting Wasserstein barycenters avoids the need to compute a Wasserstein embedding (Courty et al., 2017), and our experiments suggest that this results in better Wasserstein barycenters approximations. Our implementation is publicly available[1].

**Contributions**  This paper introduces a method to compute Wasserstein barycenters in milliseconds. It shows that this can be done by learning Wasserstein barycenters of only two measures on a dataset of random shapes using a DCNN, and by adapting this DCNN to handle multiple input

---

[1] https://github.com/iclr2021-anonymous-author/learning-to-generate-wasserstein-barycenters

measures without retraining. This proposed approach is 60x faster than the fastest state-of-the-art GPU library, and performs better than Wasserstein embeddings.

## 2 RELATED WORK

### 2.1 WASSERSTEIN DISTANCES AND APPROXIMATIONS

Optimal transport seeks the best way to warp a given probability measure $\mu_0$ to form another given probability measure $\mu_1$ by minimizing the total cost of moving individual "particles of earth". We restrict our description to discrete distributions. In this setting, finding the optimal transport between two probability measures is often achieved by solving a large linear program (Kantorovich, 1942) – more details on this theory and numerical tools can be found in the book of Peyré et al. (2019). This minimization results in the so-called *Wasserstein distance*, the mathematical distance defined by the total cost of reshaping $\mu_0$ to $\mu_1$. This distance can be used to compare probability distributions, in particular in a machine learning context. It also results in a *transport plan*, a matrix $P(x, y)$ representing the amount of mass of $\mu_0$ traveling from location $x$ in $\mu_0$ towards location $y$ in $\mu_1$.

However, the Wasserstein distance is notoriously difficult to compute – the corresponding linear program is huge, and dedicated solvers typically solve this problem in $\mathcal{O}(N^3 \log N)$, with $N$ the size of the input measures discretization. Recently, numerous approaches have attempted to approximate Wasserstein distances. One of the most efficient methods, the so-called Sinkhorn algorithm introduces an entropic regularization, allowing to compute such distances by iteratively performing fast matrix-vector multiplications (Cuturi, 2013) or convolutions in the case of regular grids (Solomon et al., 2015). However, this comes at the expense of smoothing the transport plan and removing guarantees regarding this mathematical distance (in particular, the regularized cost $W_\epsilon(\mu_0, \mu_0) \neq 0$). These issues are addressed by *Sinkhorn divergences* (Feydy et al., 2018; Genevay et al., 2017). This approach symmetrizes the entropy-regularized optimal transport distance, adding guarantees on this divergence (now, the cost $S_\epsilon(\mu_0, \mu_0) = 0$ by construction, though triangular inequality still does not hold) but also effectively reducing blur, while maintaining a relatively fast numerical algorithm. They show that this divergence interpolates between optimal transport distances and Maximum Mean Discrepancies. Sinkhorn divergences are implemented in the *GeomLoss* library (Feydy, 2019), relying on a specific computational scheme on the GPU (Feydy et al., 2019; 2018; Schmitzer, 2019) and constitutes the state-of-the-art in term of speed and approximation of optimal transport-like distances.

### 2.2 WASSERSTEIN BARYCENTERS

The Wasserstein barycenter of a set of probability measures corresponds to the Fréchet mean of these measures under the Wasserstein distance (i.e., a weighted mean under the Wasserstein metric). Wasserstein barycenters allow to interpolate between two or more probability measures by warping these measures (contrarily to Euclidean barycenters that blends them). Similarly to Wasserstein distances, Wasserstein barycenters are very expensive to compute. An entropy-regularized approach based on Sinkhorn-like iterations also allows to efficiently compute blurred Wasserstein barycenters. Reducing blur via Sinkhorn divergences is also doable, but does not benefit from a very fast Sinkhorn-like algorithm: a weighted sum of Sinkhorn divergences needs to be iteratively minimized, which adds significant computational cost. In our approach, we rely on Sinkhorn divergence-based barycenters to feed training data to a Deep Convolutional Neural Network, and aim at speeding up this approach. Other fast transport-based barycenters include that of sliced and Radon Wasserstein barycenters, obtained via Wasserstein barycenters on 1-d projections (Bonneel et al., 2015), which we compare to.

A recent trend seeks linearizations or Euclidean embeddings of optimal transport problems. Notably, Nader & Guennebaud (2018) approximate Wasserstein barycenters by first solving an optimal transport map between a uniform measure towards $n$ input measures, and then linearly combining Monge maps. This allows for efficient computations – typically of the order of half a second for 512x512 images. A similar approach is taken within the documentation of the GeomLoss library (Feydy, 2019)[2], where a single step of a gradient descent initialized with a uniform distribution is used,

---

[2]See `https://www.kernel-operations.io/geomloss/_auto_examples/optimal_transport/plot_wasserstein_barycenters_2D.html`

which effectively corresponds to such linearization. We use this technique in our work to train our network. Wang et al. (2013), Moosmüller & Cloninger (2020) and Mérigot et al. (2020) use a similar linearization, possibly using a non-uniform reference measure, with theoretical guarantees on the distorsion introduced by the embedding. Instead of explicitly building an embedding via Monge maps, such an embedding can be learned. Courty et al. (2017) propose a siamese neural network architecture to learn an embedding in which the Euclidean distance approximates the Wasserstein distance. Wasserstein barycenters can then be approximated by interpolating within the Euclidean embedding, without requiring explicit computations of transport plans. They show accurate barycenters on a number of datasets of low resolution ($28 \times 28$). However, in general, it is unclear whether Wasserstein metrics embed into Euclidean spaces. Negative results were shown for 3d optimal transport onto a Euclidean space (Andoni et al., 2016). Interestingly, in the reversed direction, Wasserstein spaces have been used to embed other metrics (Frogner et al., 2019).

Wasserstein barycenters can also be seen as a particular instance of inverse problem. There is an important literature on the resolution of inverse problems with deep learning models on instances such as (non-exhaustive list) image denoising (Ulyanov et al., 2018) (Burger et al., 2012) (Lefkimmiatis, 2017), super-resolution (Ledig et al., 2017), (Tai et al., 2017), (Lai et al., 2017), inpainting (Yeh et al., 2017) (Xie et al., 2012), (Liu et al., 2018).

Parallel to our work, Fan et al. (2020) propose a model based on input convex neural networks (ICNN) developed by Amos et al. (2017). Their method allows a fast approximation of Wasserstein barycenters of continuous input measures. This last work is also closely related to the semi-discrete approach of Claici et al. (2018).

## 2.3 Applications to machine learning

For its ability to compare probability measures, optimal transport has seen important success in machine learning. This is particularly the case of Wasserstein GANs (Arjovsky et al., 2017) that compute a very efficient approximation of Wasserstein distances as a loss for generative adversarial models. The optimal transport loss has also been used in the context of dictionary learning (Rolet et al., 2016). Other fast approximations have allowed to perform domain adaptation for transfer learning of a classifier, by advecting samples via a computed transport plan (Courty et al., 2014). Among these approximations, Sliced optimal transport has been used to sample an embedding learned by an auto-encoder, by computing a flow between uniformly random samples and the image of encoded inputs (Liutkus et al., 2019).

Regarding the Wasserstein barycenters we are interested in, they have been used for the task of learning a dictionary out of a set of probability measures (Schmitz et al., 2018), for computing Wasserstein barycentric coordinates of probability measures (Bonneel et al., 2016) or for metric learning (Heitz et al., 2019). These have been performed by automatic-differentiation of Wasserstein barycenters obtained through Sinkhorn iterations and non-linear optimization, and have thus been limited to small datasets, both due to speed and memory limitations. An adaptation of k-means clustering for optimal transport was proposed by Mi et al. (2018) and (Domazakis et al., 2020). Backhoff-Veraguas et al. replaces maximum a posteriori (MAP) estimation or Bayesian model average, by computing Wasserstein barycenters of posterior distributions (Backhoff-Veraguas et al., 2018) using a stochastic gradient descent scheme. In the context of reinforcement learning, Wasserstein barycenters are used by Metelli et al. (2019) as a way to regularize the update rule and offer robustness to uncertainty. PCA in the Wasserstein space require the ability to compute Wasserstein barycenters ; they have been studied by Bigot et al. (2017) but could only be computed in 1-d where theory is simpler. In the work of Dognin et al. (2019), Wasserstein barycenters are used for model ensembling, i.e., averaging the predictions of several models to build a more robust model.

In this work, we do not focus on a single application but instead provide the tools to efficiently approximate Wasserstein barycenters on 2-d regular grids.

## 3 Learning Wasserstein barycenters

This section describes our neural network and our proposed solution to train it in a scalable way.

### 3.1 PROPOSED MODEL

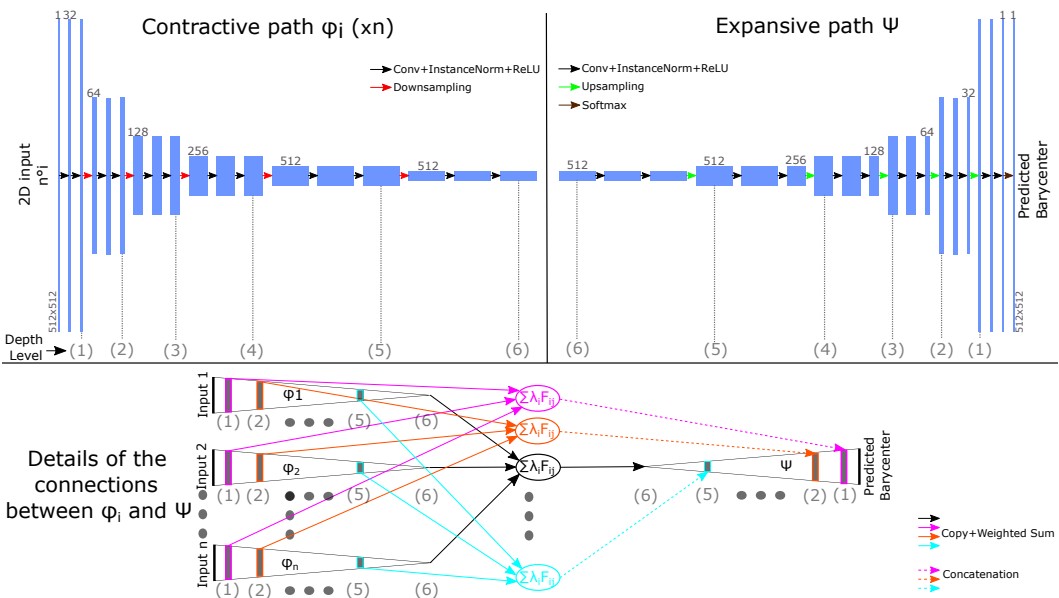

Figure 1: Our model is divided into $n$ contractive paths $\varphi_i$, sharing the same architecture and weights, and one expansive path $\psi$. Blue rectangles represent feature maps and arrows denote the different operations we use (see legend). At training time $n = 2$, but by duplicating the contractive paths, we can adapt to the $n$ measures barycenter problem at test time, without needing to retrain the network.

Our model aims at obtaining approximations of Wasserstein barycenters from $n \geq 2$ probability measures $\{\mu_i\}_{i=1..n}$ discretized on $512 \times 512$ regular grids, and their corresponding barycentric weights $\{\lambda_i\}_{i=1..n}$. Based on the observation that the Sinkhorn algorithm is mainly made of successive convolutions, we propose to directly predict a Wasserstein barycenter through an end-to-end neural network approach, using a Deep Convolutional Neural Network (DCNN) architecture. This DCNN should be deep enough to allow accurate approximations but shallow enough to reduce its computational requirements.

We propose a network consisting of $n$ contractive paths $\{\varphi_i\}_{i=1..n}$ and one expansive path $\psi$ (see Fig. 1). Importantly enough, $n$ is not fixed and can vary at test time. In fact, it is $n$ duplicates of the same path with the same architecture and sharing the same weights. The $n$ contractive paths are made of successive blocks, each block consisting of two convolutional layers followed by a ReLU activation. We further add average pooling layers between each block in order to decrease the dimensionality. The expansive path is symetrically constructed, each block also being made of 2 convolutional layers with ReLU activations. To better invert average poolings, we use upsampling layers with nearest-neighbor interpolation. Finally, to recover an output probability distribution, we use a softmax activation at the end of the expansive path. All the 2D convolutions of our model use $3 \times 3$ kernels with a stride and a padding equal to 1. The architecture might look similar to the U-Net architecture introduced by Ronneberger et al. (2015), because of the nature of the contractive and expansive paths. However the similarities end here, since our architecture uses a variable number of contractive paths to handle multiple inputs. The connections we use from the contractive paths to the expansive path also highly differ: first, we take all the feature maps from each contractive path and not only a part of it as it is done in U-Net, and, second, we compute a weighted sum of all these activations using barycentric weights which results in a weighted feature map which is then symmetrically concatenate to the corresponding activations in the expansive path. Our network is deeper than U-Net and we do not use the same succession of layers nor the same downsampling and upsampling methods which are respectively max-pooling and up-convolutions in the case of U-Net. We also use Instance Normalization (Ulyanov et al., 2016) which has empirically shown better results than Batch Normalization for our model. These normalization layers are placed before each ReLU activation.

The connections going from the contractive paths to the expansive path are defined as follows: after each block in a contractive path $\varphi_i$ at depth level $j$, we take the resulting activations $\{F_{ij}\}_{i=1..n}$, compute their linear combination $F'_j = \sum_{i \in n} \lambda_i F_{ij}$, and concatenate it symmetrically to the corresponding activations in the expansive path (see figure 1).

## 3.2 TRAINING

Our solution allows to generalize a network trained for computing the barycenter of two measures to an arbitrary number of input measures while remaining fast to train.

**Variable number of inputs.** We expect our network to produce accurate results without constructing an explicit embedding whose existence remains uncertain (Andoni et al., 2008). However, a Euclidean embedding trivially generalizes to an arbitrary number of input measures. A key insight to our work is that, since contractive paths weights are shared, our network can be trained using only two contractive paths for the task of predicting Wasserstein barycenters of two probability measures. Once trained, contractive paths can be duplicated to the desired number of input measures. In practice, we found this procedure to yield accurate barycenters (see Sec. 4).

**Loss function.** Training the network requires comparing the predicted Wasserstein barycenter to a groundtruth Wasserstein barycenter. Ideally, such comparison should be performed via an optimal transport cost – those are ideal to compare probability distributions. However, computing optimal transport costs on large training datasets would be intractable. Instead, we resort to a Kullback-Leibler divergence between the output distribution and the desired barycenter.

**Optimizer.** To optimize the model parameters, we use a stochastic gradient descent with warm restarts (SGDR) (Loshchilov & Hutter, 2016). The exact learning rate schedule we used for our models is shown in appendix A, Fig. 10.

**Training data.** We strive to train our network with datasets that would cover a wide range of input sketches. To achieve this, we built a dataset made of $100k$ pairs of $512 \times 512$ random shape contours with random barycentric weights and their corresponding 2D Wasserstein barycenter. These 2D shapes are generated in a Constructive Solid Geometry fashion: we randomly assemble primitives shapes using logical operators and detect contours in post-processing. A primitive corresponds to a filled ellipse, triangle, rectangle or a line. We assemble these primitives together by using the classical boolean operators OR, AND, XOR, NOT. To generate a shape, we initialize it with a random primitive. Then we combine it with another random primitive using a randomly chosen operator, and repeat this operation $d$ times ($0 \leq d \leq 50$) where $d$ follows the probability distribution $d \sim \frac{1}{3}(\mathcal{U}(0, 50) + \mathcal{N}(0, 2.5) + \mathcal{N}(50, 2.5))$ which promotes simple ($d$ close to 0) and complex ($d$ close to 50) shapes. Finally, we apply a Sobel filter to create contours. We thus create $10k$ random 2D shapes from which we build $100k$ Wasserstein barycenters.

We then use the GeomLoss library (Feydy, 2019) to build good approximations of Wasserstein barycenters in a reasonable time, with random pairs of inputs sampled from the set of generated shape contours. Given two 2D input distributions $\mu_1$ and $\mu_2$ with their corresponding barycentric weights $\lambda_1$ and $\lambda_2 = 1 - \lambda_1$, their barycenter $b^*$ can be found by minimizing: $b^* = \arg\min_b \lambda_1 S_\epsilon(b, \mu_1) + \lambda_2 S_\epsilon(b, \mu_2)$ where $S_\epsilon$ corresponds to the Sinkhorn divergence with quadratic ground metric, and $\epsilon$ the regularization parameter (we use $\epsilon = 1\mathrm{e}{-4}$). We use a Lagrangian gradient descent scheme that first samples the distributions as $b = \sum_{j=1}^N b_j \delta_{x_j}$ and then perform a descent using $x_j^{(k+1)} = x_j^{(k)} + \lambda_1 v_j^{\mu_1} + \lambda_2 v_j^{\mu_2}$ where $v_j^{\mu_i}$ is the displacement vector. This vector is computed as the gradient of the Sinkhorn divergence: $v_j^{\mu_i} = -\frac{1}{b_j} \nabla_{x_j} S_{\epsilon, p}(b, \mu_i)$. These successive updates can be computationally expensive when inputs are large. To speed up computations, we use a linearized approach that performs a single descent step, starting from a uniform distribution. In practice, this allows to precompute one optimal transport map between a uniform distribution and each of the input measures in the database, and obtain approximate Wasserstein barycenters by using a weighted average of these transport maps.

## 4 EXPERIMENTAL RESULTS

While our model is exclusively trained on our synthetic dataset, at test time we also consider the *Quick, Draw!* dataset from Google (2020) and the *Coil20* dataset (Nane et al., 1996). The first dataset contains 50 million grayscale drawings divided in multiple classes and has been created by asking users to draw with a mouse a given object in a limited time. The second one is made of images of 20 objects rotating on a black background and contains 72 images per object for a total of 1440 images. We rasterized these two datasets to $512 \times 512$ images.

### 4.1 TWO-WAY INTERPOLATION RESULTS

In Fig. 2, we show a visual comparison between barycenters obtained with Geomloss and our method. Wassertein barycenters are taken from the test dataset and the corresponding predictions are shown. We also compare these results to classical approaches (linear program, regularized barycenters) and to another approximation method known as Radon barycenters (Bonneel et al., 2015).

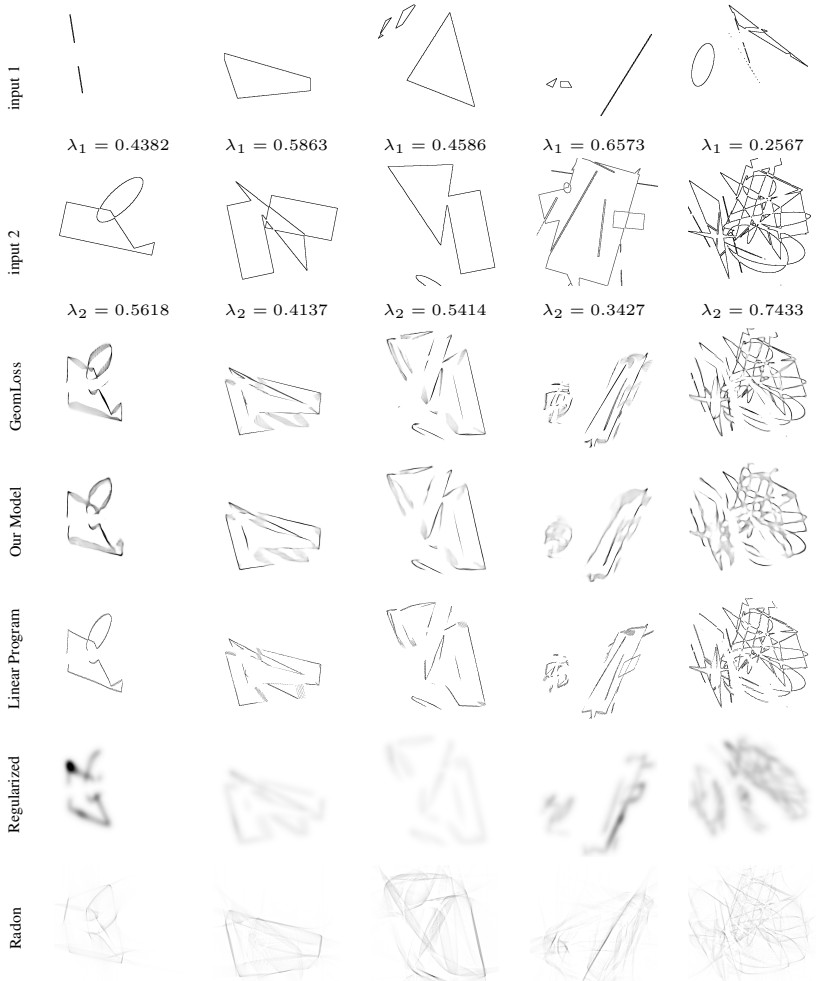

Figure 2: We illustrate typical results and comparisons to GeomLoss (Feydy, 2019), a linear program via a network simplex (Bonneel et al., 2011), regularized barycenters computed in log-domain (see for instance Peyré et al. (2019)) with a regularization parameter of $1e-3$ and Radon barycenters (Bonneel et al., 2015).

To further visually assess that the barycenters we are approximating are close to the exact ones, we also present a comparison with the method of Claici et al. (2018) in Fig. 3. Input distributions are taken from the *Quick, Draw!* dataset.

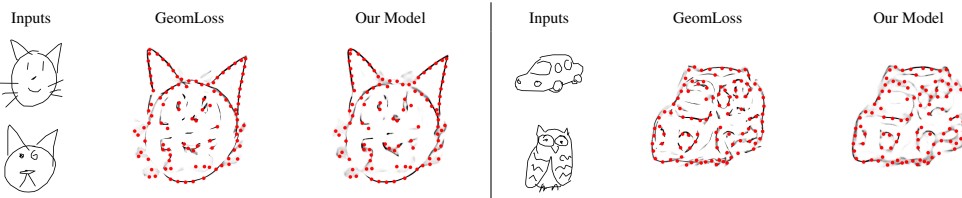

Figure 3: We superimpose the centroids ($\lambda_1 = \lambda_2 = 0.5$) found by the method of Claici et al. (2018) (in red) using 100 Dirac masses over the ones computed by GeomLoss and by our model, on images from the *Quick, Draw!* dataset. The solution of Claici et al. (2018) was found within 37 hours of computation.

We compare our method with the Deep Wasserstein Embedding (DWE) model developed by Courty et al. (2017) on *Quick, Draw!* images. We propose two versions of DWE. The first version relies on the exact original architecture which can only process $28 \times 28$ images, retrained on a downsampled version of our shape contours dataset – see Fig. 5 for this comparison. In the second version, we adapt their network to process $512 \times 512$ inputs. The encoder and decoder of this second version have the same architecture as the contractive and expansive paths that we use in our model without our skip connections, but is used to compute the embedding rather than directly predicting barycenters – see Fig. 6.

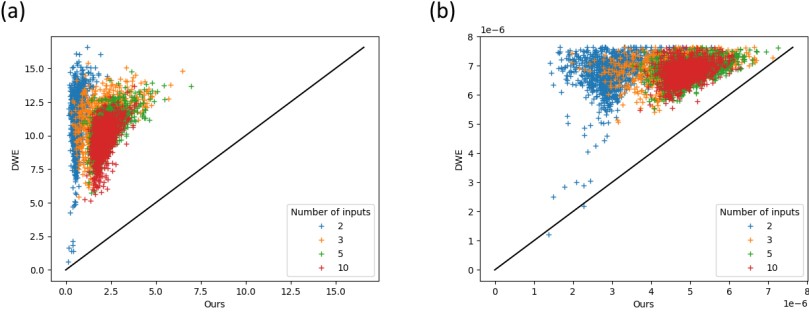

Figure 4: Approximation error of our model compared to the ones of DWE (version adapted to handle $512 \times 512$ images), respectively measured in terms of (a) KL-Divergence and (b) L1 distance, on images coming from our synthetic test dataset. Each one of the $1000 \times 4$ points corresponds to a barycenter. The x-axis represents the error measured between the GeomLoss barycenter and the barycenter predicted by our model while the y-axis represents the one between the GeomLoss barycenter and the barycenter predicted by DWE. The color of a point associated to a barycenter represents its number of inputs.

In Fig.4, we show a numerical comparison of approximation errors between our model and DWE adapted to $512 \times 512$ images of our shape contours dataset, in terms of KL-divergence and L1 distance. Our results clearly show that our method is able to approximate more accurately the Wasserstein barycenter on $512 \times 512$ input measures.

Finally, we study the limitations of the generalization of our network on the *Coil20* dataset (Nane et al., 1996), which consists of images of objects on a black background. In figure 7, we show the interpolation of 2 cars ; additional results are available in appendix B, figure 11.

## 4.2 N-WAY BARYCENTERS

Even if our model has been trained using only barycenters computed from pairs of inputs, we can apply it to predict barycenters of more than two measures. We display interpolations between respectively three and five input measures in Fig.8, which surprisingly tends to show that our model can generalize what it learned on pairs of inputs, at least partially. Additional results on *Quick, Draw!* are also shown in appendix B, Fig. 12. A 100-way barycenter comparison can be found in appendix Sec. B, Fig. 13.

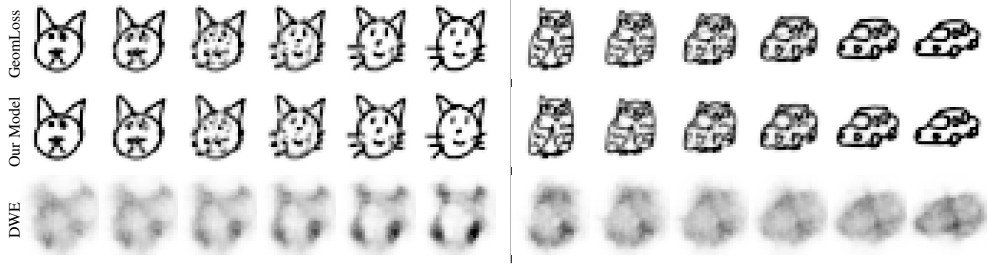

Figure 5: Interpolations between two $28 \times 28$ images from the *Quick, Draw!* dataset using Geomloss, our model and the original Deep Wasserstein Embedding (DWE) method from (Courty et al., 2017). Our model directly considers $512 \times 512$ inputs and its results are downsampled from $512 \times 512$ to $28 \times 28$.

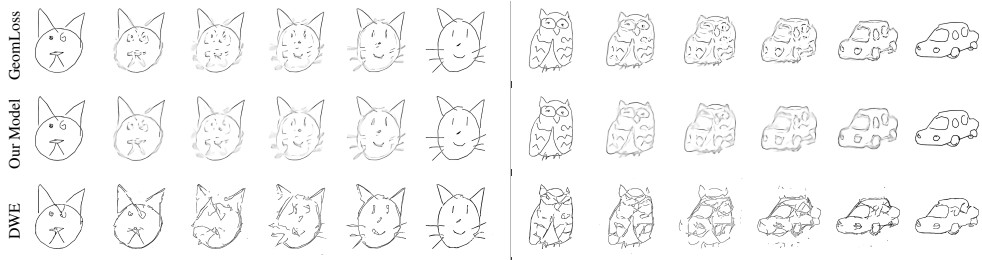

Figure 6: Interpolations between two $512 \times 512$ images from the *Quick, Draw!* dataset using Geomloss, our model and the Deep Wasserstein Embedding (DWE) method from (Courty et al., 2017) adapted to handle $512 \times 512$ images.

Numerically when the number of inputs is greater than 2, our model also achieve to find better approximations than the ones obtained with DWE, as shown in Fig. 4.

### 4.3 SPEED

In order to assess computational times, we obtain average running time over 1000 barycenter computations – on average, our model predicts barycenters of two images in 0.0092 seconds. We compare the average speed of our model with GeomLoss in two different settings. The first one considers the full $512 \times 512$ images – GeomLoss computes such barycenters in 1.41 seconds. The second setting takes advantage of the sparsity of our images and only uses the 2D coordinates of the points with non-zero mass – in this case, GeomLoss computes barycenters in 0.589 seconds. Our method provides nearly 64x speedup compared with this last approach. In comparison, an exact barycenter computation of two (sparse) measures using a network simplex (Bonneel et al., 2011) ranges from 4–80 seconds for typical shape contours images that contains few thousands of pixels carrying mass. The time required to compute barycenters using the method of (Claici et al., 2018) depends on the number of iterations, in our setting 100 iterations with the inputs shown in figure 3 require 37 hours while 50 iterations are achieved in 14 hours. A $512 \times 512$ Radon barycenter (Bonneel et al., 2015) requires 0.2 seconds for 720 projection directions, but remains far from the expected barycenter.

### 5 DISCUSSION AND CONCLUSION

While our method produces good approximation of Wasserstein barycenters of $n$ inputs, some shapes are surprisingly difficult to handle. The barycenter of simple translated and scaled shapes such as lines or ellipses should theoretically also be lines or ellipses, but are failure cases for our model (Fig. 9), while more complex shapes are well handled (Fig. 8). In addition, we rely on a linearized barycenter to train our network (Nader & Guennebaud, 2018; Wang et al., 2013; Moosmüller & Cloninger, 2020; Mérigot et al., 2020), which incurs some error. This can be seen in appendix Sec. C, Fig. 14. While using more iterations of gradient descent yields more accurate results and removes this linearity, it also prevents easy combination and makes the dataset generation intractable.

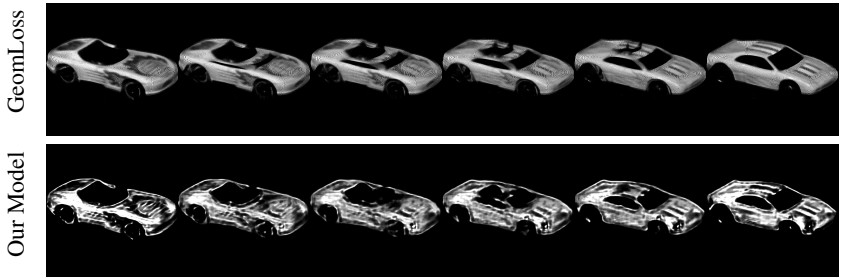

Figure 7: Interpolations between two $512 \times 512$ images from the *Coil20* dataset (Nane et al., 1996) using GeomLoss and our model trained with synthetic shape contours. In order to perform computations on the shapes and not on the background, mass has been inverted.

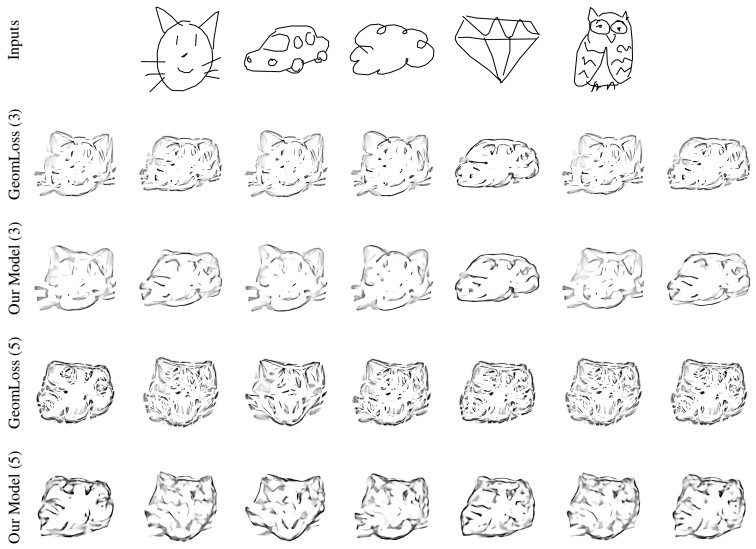

Figure 8: Wasserstein barycenters of three inputs (top rows) and five inputs (bottom rows) from *Quick, Draw!*, respectively computed with Geomloss and with our model trained with only pairs from our synthetic training dataset. Barycentric weights are randomly chosen.

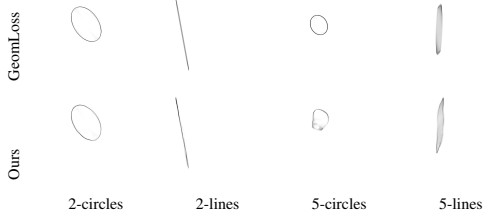

Figure 9: Wasserstein barycenters of sets of lines or ellipses should result in lines (resp. ellipses). Our prediction for two-way barycenters (here, with equal weights) of such shapes remains correct (left). However, the predicted barycenter is highly distorted for 5-way barycenters of simple shapes (right) although it remains plausible for more complex shapes (see Fig. 8).

Nevertheless, in many cases our DCNN is able to synthesize a barycenter from an arbitrary number of inputs. The main strength of our approach lies in its capacity to be trained from only 2-inputs barycenters examples and to generalize to any number of inputs. We showed that the results exceeded the ones obtained by explicit Wasserstein Embedding computation while having a very low computation time. We hope our fast approach will accelerate the adoption of optimal transport in machine learning applications.

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

## A    LEARNING STRATEGY

Instead of using a fixed learning rate or a decreasing learning rate, we choose a learning rate schedule with warm restart as proposed by Loshchilov & Hutter (2016). The learning schedule is shown in Figure 10: the learning rate decreased and is periodically restarted to its initial value, the period increasing as the number of epochs grows. This schedule was chosen after comparing with stepwise schedules or constant learning rates and yielded better convergence in practice.

## B    ADDITIONAL RESULTS

To better show the limitations of the generalization of our network when the number of inputs is 2, we show additional interpolations between 2 objects from the *Coil20* in figure 11. There are two reasons for these bad results: first, our model is trained in synthetic shape contours and do not look at all like these images. Furthermore, the cup image seem to be even more challenging than the car image for our network, and our best explanation for this failure is that the cup covers almost the whole image. We provide additional experiments showing barycenters of 5 sketches on Figure 12. The weights evolve linearly inside the pentagon. As a stress test, we also show a barycenter of 100 cats with equal weights in Fig. 13 and compare it with a barycenter computed with GeomLoss. While both results recover more or less the global shape of the cat, details are clearly lost and our result looks much smoother.

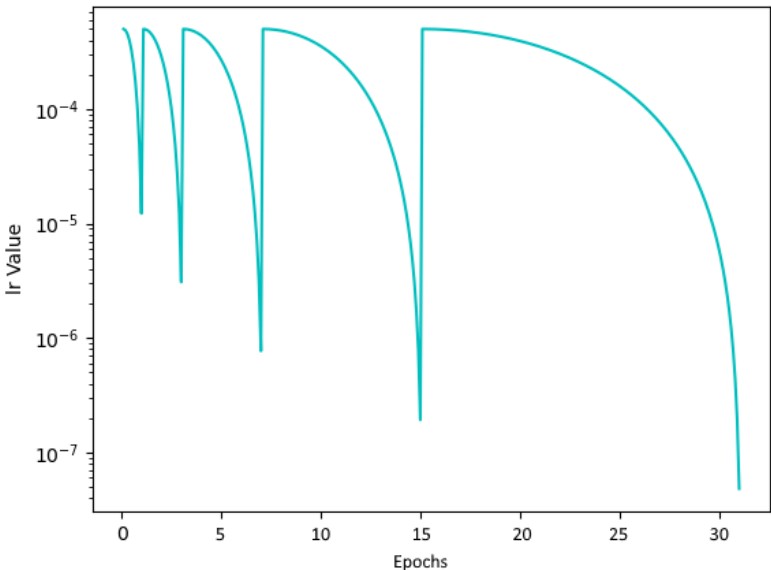

Figure 10: Learning rate schedule used to train our models, following the SGDR method described by Loshchilov & Hutter (2016). Our training runs for a total of 31 epochs. Compared to a constant learning rate or to stepwise schedules, SGDR has empirically shown a better convergence in our context.

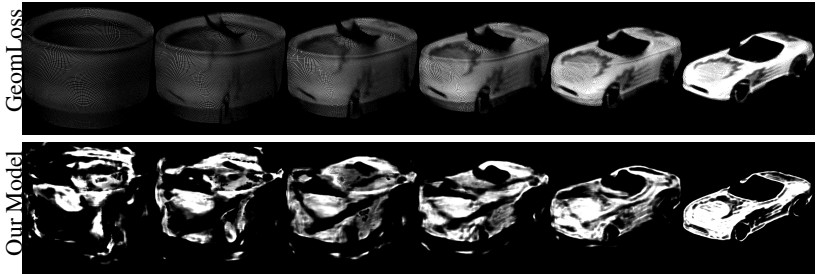

Figure 11: Additional interpolations between two $512 \times 512$ images from the *Coil20* dataset using GeomLoss and our model. Mass is inverted.

## C  LINEARIZED BARYCENTERS

Fig. 14 shows the error introduced by using a linearized version of Wasserstein barycenters (Nader & Guennebaud, 2018; Wang et al., 2013; Moosmüller & Cloninger, 2020; Mérigot et al., 2020). Our predicted barycenters reflect this error.

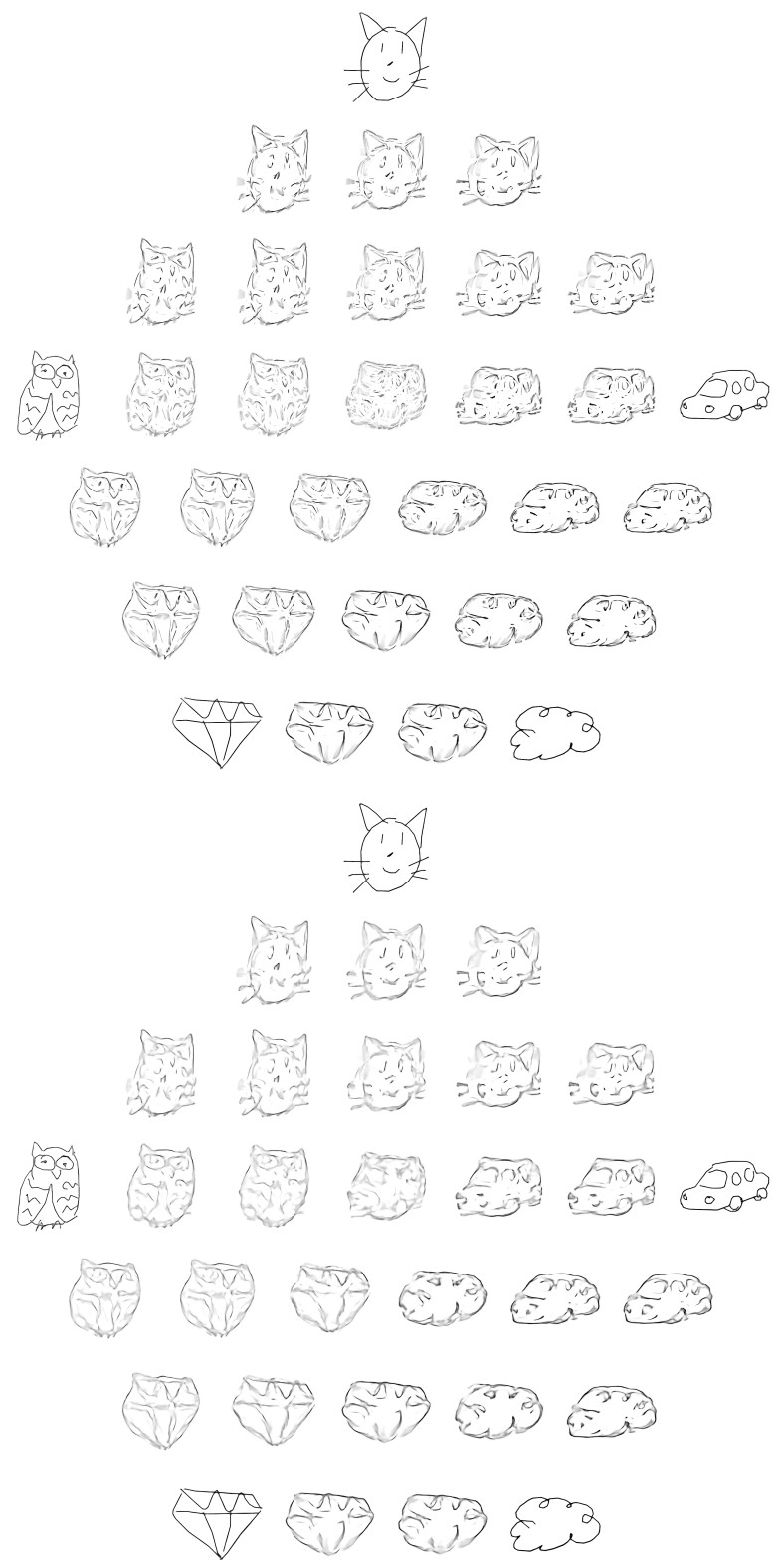

Figure 12: Interpolations between 5 inputs from *Quick, Draw!*, shown as pentagons. Left pentagon corresponds to GeomLoss barycenters while the right one shows predictions of our model trained on our synthetic dataset.

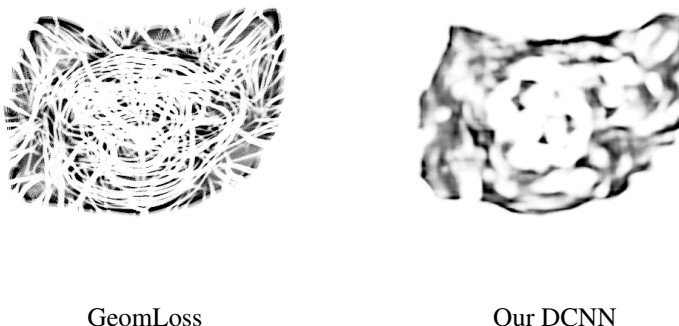

GeomLoss                                Our DCNN

Figure 13: Stress test. We predict a barycenter of 100 cats of the QuickDraw dataset, with equal weights.

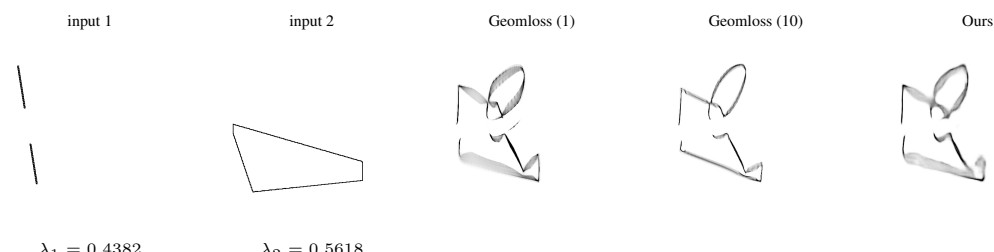

Figure 14: Wasserstein barycenter computed from a pair of inputs respectively using Geomloss with only one descent step, Geomloss with 10 descent steps and using our model trained on our synthetic training dataset.

