# OpenReview forum: "Learning to generate Wasserstein barycenters"
_ICLR.cc/2021/Conference — Reject_

### Official Review · AnonReviewer3 · 2020-10-27
**Limited contributions**

**Rating:** 3
**Confidence:** 4

**Review:**

This paper proposes a numerical method to approximate 2d Wasserstein barycenters, that relies on a convolutional network. Namely, the proposed architecture is composed of
1.	Contractive paths, made of series of 2 convolutional layers, relu activation and a pooling step, characterized by weight sharing.
2.	An expansive path, symmetrically constructed, with upsampling by NN interpolation, and a final softmax activation.

Though fixed during training, the varying number of contracting paths in the test setting is able to predict the barycenter of a varying number of measures. The KL loss is used to train the network predicting the Wasserstein barycenter.
The method is assessed on the Quick-draw dataset, to produce barycenters of two measures and its generalization to barycenters of n measures.

Strong points of the paper include:
1.	The writing of the paper is clear.
2.	The goal to propose a speedup for the computation of Wasserstein barycenters with a varying number of input measures is pertinent.

Weak points include:
1.	The paper proposes an architecture tailored for very specific setting of 2d images. Its extensions to higher dimensional or non-image settings are limited and are not discussed.
2.	Though mentioned as closely related works, the paper is missing important baselines, for instance (Claici et al.,2018).
3.	The fact that only one dataset is considered undermines the paper, which focuses on numerics.
4.	The figure at the end of page 6 (which is lacking both title and legend) does not provide a relevant metric to assess the benefits of the proposed method.
5.	The paper does not provide other metrics (such as inception scores) to assess the performance of the methods other than visually.

I recommend a reject on the aforementioned grounds.

---

> ### Author Response · Authors · 2020-11-17
> **Response to AnonReviewer3**
>
> We thank the reviewer for its constructive feedback, and detail below our answer.
> 1. While we did not consider non-image settings in our paper, we expect our model to be trivially extended to voxel grids.  However, the unstructured data case is much more complicated since it cannot benefit from the fast convolutions of our deep convolutional network.
> 2. We thank the reviewer for this suggestion and we now provide a comparison with (Claici et al., 2018) in part 4, figure 3. However note that this method is extremely slow: a typical run time we achieved using Claici et al.’s public implementation was 14 hours of computation for a barycenter of two measures consisting of 50 Dirac masses, and 37 hours for 100 Dirac masses when using 2 images from the QuickDraw dataset (while our method works with images that contains 2621 times more data and produces results in milliseconds). This gives the opportunity to emphasize the difficulty of the problem we are tackling.
> 3. Only the synthetic dataset is considered for the training of our model ; however we provide examples of results on different datasets : Quick Draw, simple shapes (circles, lines) and to answer to the concerns of the reviewer we have now also added comparisons with the COIL20 dataset.
> 4. The figure which was at the end of page 6 now corresponds to figure 4. It has been revised and we provide additional results inside it in order to allow a comparison with the DWE model for a number of input distributions greater than 2. Note that this visualization is standard in data analysis, and has been notably used by [Courty et al. 2017] which we compare to.
> 5. While inception scores do not seem relevant in our context as, to the best of our knowledge, they are tailored for GANs, the advice of the reviewer to provide other metrics is fair. We thus provide an additional figure together with the figure 4, which measures errors in terms of L1 distance.

---

### Official Review · AnonReviewer4 · 2020-10-28
**A modification of U-Net that computes approximate Wasserstein barycenters**

**Rating:** 7
**Confidence:** 3

**Review:**

A modification of U-Net to compute Wasserstein barycenters

This paper presents a deep network architecture which learns to generate Wasserstein barycenters associated with a given set of input measures. The paper is clearly written and presents a fairly objective view on their work. It gives one more example of optimization task (here the convex minimization problem of Wasserstein barycenters) that can be replaced by deep neural networks with seemingly satisfactory results.
Overall, although the method is relatively straightforward, it has the merit of showing that fast approximations of linear approximations of Wasserstein barycenters can be performed at a cheap forward pass cost of a U-Net. Note that the method has no built-in information on the structure of the problem e.g. preservation of total mass for instance or more higher-level properties.  These fast approximations could be proven to be useful in other settings outside machine learning too.

The benefit of the proposed architecture is that it can be adapted to compute the barycenters for any given number N of inputs while only trained on pairs (N =2) of input measures. Apart from minor modifications in the pipeline, the architecture remains close to that of a U-Net with the modification that the barycenters weights are explicitly used to compute averages of the activations at different layers of the Deep CNN. This work is quite weak in terms of novelty (structure of the network for instance).

The experiments are compared against and trained with the GeomLoss code which seems to be state of the art for computing entropic regularity. They also compare with Wasserstein embeddings methods favorably.

— References to literature in using deep learning for solving inverse problems would be welcome. Wasserstein barycenter is a particular instance of such class.

— Comments on the sensitivity of the results to training data would be welcome.

— I find the description of the algorithm for building the training data quite unclear. If I understand correctly, the training data is an approximation of Wasserstein barycenters between two measures. These approximations are simply computed with one step of the gradient descent of the Sinkhorn divergence and use an average of the corresponding transport maps. The cost of computation of these approximate barycenters are essentially two sinkhorn loops up to the given tolerance. This would have been interesting to have a comparison, in the supplementary material, of the difference between these approximations and more accurate approximations of Wasserstein barycenters such as given by entropic regularization with very small regularization parameter and discuss in more details what can be the impact of such inaccuracies. The authors say « While using more iterations of gradient descent yields more accurate results and removes this linearity, it also prevents easy combination and makes the dataset generation intractable ». I do not understand this statement. Cost of the gradient descent linearly scales with the number of steps, so what do the authors mean by « easy combination », the linear approximation step ?

Minor remarks:
— the (not numbered) figure on page 6 called adjacent is almost impossible to read and it is not explained enough.
—  A dot is missing after KL-divergence on page 6.
— Why is DWE performing so badly in figure 3 in comparison with figure 4 ?

---

> ### Author Response · Authors · 2020-11-17
> **Response to AnonReviewer4**
>
> We thank the reviewer for its constructive feedback, and detail below our answer.
> - As suggested, we added references to literature using deep learning for solving inverse problems in part 2.2.
> - In order to further show the generalization ability and the limitations of our model, we provide additional comparisons on the COIL20 dataset in part 4 and in the corresponding figures 7 and 11.
> - As written in our answer to AnonReviewer1, our method computes approximate Wasserstein barycenters learnt from a dataset of approximate Wasserstein barycenters computed with Sinkhorn divergences which were parameterized with a small regularization parameter of 1e-4 (the value of the regularization parameter was incorrect in the initial submission and it has now been corrected). Sinkhorn divergences are meant to be an improvement over classic regularization methods and are thus expected to provide sharper Wasserstein barycenters. We added in figure 2 a visual comparison with entropic regularized Wasserstein barycenters, computed using the Sinkhorn algorithm with log-domain computations (as described e.g. in Peyré & Cuturi’s book).
> About “easy combinations”: for each image of our dataset, we precompute one approximate optimal transport map between a uniform distribution and the image itself. We can then combine $n$ pre-computed optimal transport maps corresponding to $n$ images in order to obtain several approximate Wasserstein barycenters at no additional cost. However, this “easy combination” is only possible when using a single gradient step and does not apply when the number of gradient steps is higher than one. If we had used 2 or more gradient steps, we would have had to compute optimal transport maps each time we compute a Wasserstein barycenter, which would be intractable for building a large training dataset.
>
> Minor Remarks:
> - The figure which was on page 6 now corresponds to figure 4 and it has been corrected and enhanced.
> - The typo has been corrected.
> - As stated in section 4.1, we consider 2 versions of DWE : the original architecture which can handle only 28x28 images and a modified architecture well suited for 512x512 images. The encoder and decoder of this last version have the same architecture as the contractive and expansive paths used in our model. In figure 5 our model takes 512x512 inputs and its results are downsampled in 28x28 images while DWE takes downsampled 28x28 inputs and returns 28x28 results. In figure 6, there is no downsampling and the 2 models directly take 512x512 inputs. We clarified the associated parts in section 4.1.

---

### Official Review · AnonReviewer1 · 2020-10-29
**An interesting experiment with Wasserstein Barycenters**

**Rating:** 6
**Confidence:** 4

**Review:**

Summary:

This paper proposes a network architecture and a training model for learning Wasserstein barycenters of 512x512 discrete probability distributions. Results are compared with other approximation methods.

Reason for my score:

This paper is ell written, and the idea is interesting.  However, the elephant in the room is: How close are these learned barycenters from the actual barycenter? For me this is the main question, that cannot be answered with CNN approaches, and thus, makes it difficult to translate into uses for optimal transport techniques. This limits the impact that the paper can have on a broader audience.

The authors made a great effort to provide a rather complete review of the literature, and the provided experiments are sound. However.

1. The comparison of the results is made against another approximating method. Again, I would like to see some evidence that the learned model is actually generating good approximations of the Wasserstein Barycenter.

2. It is a less exciting if the loss function used is not OT itself but KL.

3. The main advantage of entropy regularized methods is that one can actually control how close the computed WB is from the actual WB. This is a main issue. In the proposed method, we have no way to know this, and I cannot avoid thinking this is just an informed guess.

4. I believe one main advantage of the proposed method would be to generate initialization points of entropy regularized approaches.

5. The proposed architecture is well explained. Why not other architecture, some experiments exploring other ways would be appreciated as well.

6. For some basic distributions it would have been more useful to compare the learned barycenter with the output of the linear program. At least in a couple of cases.  This can provide some evidence that the learned barycenter is actually a good approximation.

---

> ### Author Response · Authors · 2020-11-17
> **Response to AnonReviewer1**
>
> We thank the reviewer for its constructive feedback, and detail below our answer.
> 1. The production of Wasserstein Barycenters via more exact methods - such as linear programs or e.g. via Claici et al.’s method - remains computationally highly expensive. We provide some visual comparisons between our approximated Wasserstein barycenters and ones generated by these more exact methods (a visual comparison with Claici et al. barycenters has been added, figure 3), but a numerical analysis would require a high number of Wasserstein barycenters produced with exact methods, which would be computationally intractable.
> A typical run time we achieved using Claici et al.’s public implementation was 14 hours of computation for a barycenter of two measures consisting of 50 Dirac masses, and 37 hours for 100 Dirac masses when using 2 images from the QuickDraw dataset.
> 2. The use of OT as a loss in our network would also have been computationally expensive and intractable. Fast OT approximations such as Sliced OT may seem an interesting replacement of KL divergence, however our preliminary results with sliced OT were not promising with our model so we did not consider it further.
> 3. Our method computes approximate Wasserstein barycenters learnt from a dataset of approximate Wasserstein barycenters computed with Sinkhorn divergences obtained with a small regularization parameter of 1e-4.  Sinkhorn divergences are meant to be an improvement over classic regularization methods and are thus expected to provide sharper Wasserstein barycenters. We added in figure 2 a visual comparison with entropic regularized wasserstein barycenters, computed using the Sinkhorn algorithm with log-domain computations (as described e.g. in Peyré & Cuturi’s book). Note that regularizing optimal transport with entropy has been shown to improve Wasserstein barycenters over exact methods such as linear programs (e.g., Cuturi & Peyré 2016, https://arxiv.org/pdf/1503.02533.pdf , Fig. 3.1 and 3.2).
> 4. We believe our method can provide initialization points to iterative methods such as the one adopted in (Claici et al., 2018). We think it could be interesting to investigate this in a future work.
> 5. Some variants of our architecture have also been considered, in particular : depth of the network, number of layers per level, downsampling with max-pooling, down-sampling and up-sampling with convolutions only, Batch Normalization instead of Instance Normalization. The architecture we present is the one which has empirically shown the best results among the considered configurations.
> 6. Unfortunately, linear programs are out of reach for problem sizes handled by our network. A typical barycenter of two 512x512 images would require storing a 512^4 matrix of floating point values, requiring 512 GB of RAM. The state-of-the-art linear program solver (see https://github.com/nbonneel/network_simplex) handles at most problems that are orders of magnitude smaller (about 50K Diracs), and would not support barycenters of more than two measures of any size.

---

### Decision · Program_Chairs · 2021-01-07
**Final Decision**

**Decision:**

Reject

**Comment:**

Given the reviewer's exchange with the authors, and my own examination of the paper, I don't think that it can be accepted in the present form.

First, since this paper aims at solving an optimization problem (for which existing methods exist, with theoretical guarantees) via a NN, it is important to compare appropriately to those methods, which is not done here.

Further, there are possible issues when applying these only to 2D data, and it is possible that it would not extend appropriately to other types of geometries, and costs in OT problems.